# Three Diagnoses for Problematic Hypersexuality; Which Criteria Predict Help-Seeking Behavior?

**DOI:** 10.3390/ijerph17186907

**Published:** 2020-09-21

**Authors:** Piet van Tuijl, Aerjen Tamminga, Gert-Jan Meerkerk, Peter Verboon, Ruslan Leontjevas, Jacques van Lankveld

**Affiliations:** 1Department of Psychology, Open Universiteit, 6419 AT Heerlen, The Netherlands; peter.verboon@ou.nl (P.V.); roeslan.leontjevas@ou.nl (R.L.); jacques.vanlankveld@ou.nl (J.v.L.); 2Psychologen Nederland (PsyNed), 1062 GD Amsterdam, The Netherlands; aerjen@psyned.nl; 3Instituut voor Onderzoek naar Leefwijzen & Verslaving (IVO), 2595 AA Den Haag, The Netherlands; meerkerk@ivo.nl

**Keywords:** sex addiction, hypersexuality, compulsive sexuality, sexual frequency, withdrawal, tolerance, coping

## Abstract

This study aimed to assess the best combination of indicators of problematic hypersexuality (PH), in a survey (*n* = 58,158) targeting individuals wondering if they were sex addicted. The survey allowed for testing of criteria from three theoretical models used to conceptualize PH. Factor analyses for women and men yielded an interpretable grouping of indicators consisting of four factors. In subsequent logistic regressions, these factors were used as predictors for experiencing the need for help for PH. The factors Negative Effects and Extreme positively predicted experiencing the need for help, with Negative Effects as the most important predictor for both women and men. This factor included, among others, withdrawal symptoms and loss of pleasure. The Sexual Desire factor negatively predicted the need for help, suggesting that for the targeted population more sexual desire leads to less PH. The Coping factor did not predict experiencing the need for help. Outcomes show that a combination of indicators from different theoretical models best indicates the presence of PH. Therefore, a measurement instrument to assess existence and severity of PH should consist of such a combination. Theoretically, this study suggests that a more comprehensive model for PH is needed, surpassing existing conceptualizations of PH.

## 1. Introduction

Problematic hypersexuality (PH) can be defined as the experience of problems due to intense and/or highly frequent sexual behavior, preoccupations, thoughts, feelings, urges, or fantasies that are out of control [1,2]. Prevalence of PH is estimated to be at least 2% of the population [3], with estimates in some subpopulations as high as 28% [4,5]. A two to three times higher prevalence in men than in women has been found [3,6]. The existence of PH and the possibility to diagnose PH are vehemently debated [7,8,9,10]. Particularly, the potentially overpathologizing effect of a diagnosis is criticized, and some characterize a clinical diagnosis for PH as merely a description of disapproved sexuality [11]. Despite the difficulties to clinically define PH, of which the diverging current diagnoses bear witness [12,13,14,15], clinicians have testified that the condition is clearly experienced by their clients [16,17,18], be it formally diagnosable or not. Due to the conceptual confusion and the lack of research, it might still be too early to define PH clinically. Thus, the working definition of PH we proposed above refers more to a behavioral complex [12] than to a formal diagnosis.

In recent years, partly conflicting theoretical models have been developed to establish PH as a clinical syndrome. Specific diagnostic criteria have been developed based on three of these models. PH is viewed as (1) sex addiction [19,20,21,22,23,24], (2) hypersexual disorder [25,26,27], or (3) compulsive sexual behavior disorder [28,29]. Sex addiction as a clinical diagnosis is characterized by generic addiction indicators, such as preoccupation, negative interference of the sexual behavior with daily activities, failure to quit, continuation despite negative consequences, tolerance, and withdrawal symptoms [23,24]. Hypersexual disorder has been proposed—and later rejected—as a diagnosis for the DSM-5. Its diagnostic model contains several of the sex addiction criteria, though not those of tolerance and withdrawal [25]. Based on influential research [30], criteria for sex used as coping [25] (criteria A2 and A3) were included as part of the hypersexual disorder diagnosis. Despite the rejection of this diagnosis for inclusion in the DSM-5 [31], a scale with items addressing coping remains part of the Hypersexual Behavior Inventory [32], a frequently used instrument to assess PH. The relatively high percentages of hypersexual individuals that have been found with this instrument [4,33] suggest that associations between coping and sexuality might also be problematic for part of the general population that is not specifically afflicted by PH. Compulsive sexual behavior disorder, the newly accepted ICD-11 diagnosis [28], differs from the sex addiction diagnosis mostly in the addition of one indicator and a set of guidelines. The indicator stresses the continuation of repetitive sexual behavior despite loss of pleasure [28]. The guidelines caution against overpathologizing, particularly of preoccupation with sex [28] and distress related to feelings of guilt and shame [29].

A number of criteria used in the three diagnostic models for PH have not been thoroughly studied. The criterium of loss of pleasure has not been quantitatively investigated at all; a high prevalence of tolerance and withdrawal symptoms was found among clinical in- and outpatients treated for sex addiction [23], but in the one study investigating this prevalence, a comparison group that was not affected by PH was not included. A similar research design problem occurs in a number of studies on sexual frequency and PH of which the results suggested that, analogous to substance addiction, higher sexual frequency predicts the occurrence of PH [34,35,36]. However, when relevant comparison groups were included in large-scale studies, higher sexual frequency did not discriminate between PH and high sexual desire without distress [37,38]. These conflicting results with regard to sexual frequency suggest that (1) a higher percentage of PH will be found in the general population among those with higher sexual frequency [36,37,38] and that (2) among those for whom it might be relevant to know if they are at risk of PH, sexual frequency might not be a discriminative indicator [39]. This does not include nor exclude high sexual frequency as part of a diagnosis for PH, but it does suggest that high sexual frequency cannot be used to discriminate PH from other, nonclinical, conditions, in particular high sexual frequency without distress.

In this exploratory research of a large-scale internet sample, a first step is taken to establish which criteria of the three different diagnostic models are unique indicators that distinguish PH from other conditions. These indicators will have high discriminative power and will lead to valid and reliable cues [40,41] to understand and assess PH. Accordingly, the most important aim of this study is to explore and test an extended set of characteristics and establish which can be used best to assess PH. For this, we make use of a sample in which relevant subgroups can be compared [42]. Furthermore, we aim to investigate if a larger number of relevant indicators present in individuals increases the probability that they will experience the need for help for PH. If this is the case, it would suggest that these indicators can be part of an instrument that not only has discriminative power but can also measure the severity of PH. With a measure of severity, evaluations of interventions might be performed and therapeutic progress might be assessed [43]. In this study, special attention will be given to gender differences as it cannot be assumed a priori that women and men experience PH in the same way.

## 2. Materials and Methods

### 2.1. Study Population

In the Netherlands, concerns about the prevalence of sex addiction [44] led to the construction of a survey on the Netherlands-based online platform for psychological help, www.sekned.nl, owned at the time of data collection by PsyNed, Psychologen Nederland (Psychologists Netherlands) and currently owned by NCVS, Nederlands Centrum Voor Seksverslaving (Dutch Center for Sex Addiction). The survey targeted those in doubt about being addicted to sex and aimed at providing participants with a preliminary self-assessment about their level of PH. As the term “sex addiction” might have had many connotations for participants, some of which include distress while others simply express a nondistressing preoccupation with sex [38], it can be expected that also those not afflicted by PH, but experiencing high sexual desire without distress, will seek information from this survey.

### 2.2. Survey and Sample

The survey underlying this research gathered responses of 58,158 participants between July 2014 and July 2018. The first objective of the survey was to provide feedback to participants on their level of PH. Before and after taking the survey, participants were notified that the data collected might also be used for scientific research. Data were not collected with a research strategy in mind, and the current research has been set up after data collection finished. As the data were classified as secondary data, the study was considered exempt from ethical approval by the ethical approval board of the Open University Netherlands. In order to secure anonymity, IP addresses were not recorded but changed into anonymous code. Completed surveys did not contain information that could be traced back to participants. Only fully completed surveys were kept for analysis, but were excluded when (1) participants belonged to the age group 17−21 or younger (*n* = 17,689) because parental approval could not be obtained, (2) participants indicated that they completed the survey for someone else (*n* = 3467), and (3) IP addresses were not used for the first time (*n* = 3842). In total, 33,160 completed surveys were included in the analyses, of which 25,733 (77.8%) were filled in by men and 7427 (22.4%) by women. In total, 7583 (22.9%) participants expressed interest in seeking help for PH. Previous analyses of this same dataset have been published in Dutch [39]; these analyses did not use the current extended research design with separate factor analyses for women and men.

### 2.3. Exploratory Nature of this Research

This research must be considered exploratory because data were collected before the research design was set up. This means that the character and number of items used in the survey could not be determined beforehand by researchers. Nonetheless, a number of relevant items pertaining to PH have been incorporated in the survey, covering criteria from the three diagnostic models for PH. With regard to the validity of the results of this study, confirmative research will be needed to further investigate the explorative conclusions. As the survey gathered a large response of participants interested in their level of addiction to sex, this sample can be considered unique to the field of PH research as it is often problematic to collect extensive data from participants afflicted by PH (e.g., [36]). By investigating a sample of participants in doubt about being addicted to sex, we limit ourselves to a subpopulation [42] for whom adequate cutoff points should be established, because it is for this group that the risk of misdiagnosis is highest and the consequences of misdiagnosis most detrimental [45]. Though there is no certainty that the investigated subpopulation consists indeed of those in doubt about their level of sex addiction, the introduction to the survey clearly stresses its purpose as providing a preliminary self-assessment of the participant’s level of sex addiction; also completion of the survey, needed to receive the feedback, shows an interest in its outcome and suggests the targeted subpopulation was reached.

### 2.4. General Indicators of Problematic Hypersexuality

A set of indicators that is part of the criteria of all three diagnostic models for PH consists of (1) preoccupation with sex (“I spend a lot of time on anything related to sex”), (2) failed attempts to quit (“I don’t succeed in stopping though I often tried”), (3) continue despite negative consequences (“I continue despite knowing it is not good for me”), and (4) occurrence of negative consequences (“My craving for sex has cost me much”). Answers on these four items are categorized as either “0 (no)” or “1 (yes)”.

### 2.5. Sex Addiction Indicators

Characteristics that are typically only used as indicators of sex addiction but not used as indicators in the other diagnostic models are (1) tolerance (“I want to have sex more and more”, answers categorized as either “yes” or “no”), and (2) withdrawal symptoms (“When I try to stop I feel nervous and restless,” scores ranging from “0 (never)” to “4 (always)”).

### 2.6. Hypersexual Disorder Indicators

Indicators that can be specifically linked to the hypersexual disorder diagnostic model concern the six items on coping in the survey (e.g., “I feel less depressed after sexual activity” or “I need sex to function well”, answers categorized as either “0 (no)” or “1 (yes)”).

### 2.7. Compulsive Sexual Behavior Disorder Indicator

Only one indicator included in the compulsive sexual behavior disorder diagnostic model addresses the continuation of sexual behavior despite loss of pleasure (“I feel empty after being sexually active”, answers categorized as either “0 (no)” or “1 (yes)”).

### 2.8. Need for Help

Two items assessed the need for help: 1) “I would like to receive individual or group therapy”, and 2) “I would like to participate in an internet-based training”. Answers were categorized as either “0(no)” or “1 (yes)”. Any affirmative answer categorizes the respondent as being part of the category “Experiencing the need for help for problems due to PH”, codes as “0 (no)” or “1 (yes)”.

### 2.9. Covariates

A set of six possibly relevant covariates was selected from the survey to be part of the analyses. These include aspects that might be related to PH but are not explicitly mentioned in the criteria for any of the three diagnostic models for PH. For most covariates, there has been research into the associations with PH. The six covariates are (1) Orgasm frequency (“I usually have had an orgasm: “0 (less than once a day)/1 (equal to or more than once a day)“) [34,35]; (2) Time spent watching pornography (“How much time per day do you spend watching porn?”, six answer categories ranging from “never” and “0 to 30 min” to “4 to 6 h”) [44]; (3) Watching more extreme pornography (“I watch more and more extreme porn: 0 (No, I don’t watch porn)/1 (No, I watch less extreme porn)/2 (No, I watch the same sort of porn)/3 (Yes, I watch more extreme porn)”; (4) Watch porn in combination with use of drugs (“I use stimulants before or during watching porn (e.g., alcohol)”, five answer categories ranging from “0 (never/I don’t watch porn)” to “4 (always)”; (5) Social pressure (“Someone has told me I should stop”, answers categorized by “0 (yes)” or “1 (no)”) [17]; and (6) Paraphilic orientation (one item: “I use unusual sexual stimuli (e.g., sex with animals or children)”, five answer categories ranging from “0 (never)” to “4 (always)”) [34]. With regard to “Orgasm frequency”, “Time spent on porn”, and “Unusual sexual stimuli” (paraphilic orientation), prior research showed some associations with PH, but these associations remained equivocal. With regard to “Watch porn while using drugs” and “Extreme porn”, there has been less research, but these two indicators would align with an escalating pattern of PH as conceptualized in the sex addiction perspective and thus were included as covariates. “Social pressure” has also not been investigated quantitatively but has been suggested as an equivocal aspect of PH by sexologists [17] that needs to be studied more. The covariates Age and Gender are also included in the analyses: Age is divided in six categories ranging from “22 to 31” to “older than 60”; Age is used as a control variable in the final logistic regression analysis (see Section 2.8). Gender (categorized as “woman” or “man”) is used in analyses to test if response patterns for women and men are similar by performing the analyses for both genders separately and comparing the results (see Section 2.10).

### 2.10. Statistical Analyses

Exploratory analyses have been designed to investigate the indicators of PH available in the collected data. Special attention was given to the factor structure of the variables; establishing which indicators belonged together allowed for a better interpretation of the indicators and also made it possible to investigate the predictive and discriminating properties of the factors. Follow-up research will be needed to confirm the explorative results of this study.

Analyses were conducted separately for women and men as response patterns for both sexes are expected to be different and it was our aim to investigate these differences. Separate analyses also avoid the risk of bias of gender. Absolute and relative frequencies or means and standard deviations of the included variables were described for four different groups: (1) women needing help, (2) men needing help, (3) women not needing help, and (4) men not needing help. Receiver operating curve analyses have been included to determine the discriminative power of each separate variable in discerning those who experience the need for help from those who do not want help for PH. The results of these analyses are area under the curve values (AUC) that provide a measure of the discriminative power of each variable, with values significantly higher than 0.5 representing indicators that, when present, can be used to assess PH. AUC values closer to 1 signify indicators with more discriminative power.

To be able to interpret the variables better, the underlying factor structure of the variables was investigated first with exploratory factor analysis (EFA) and then with confirmatory factor analysis (CFA). EFA was performed to establish the number of factors. A limited and randomly selected part of the data was used to perform analyses, with separate EFAs for women (*n* = 1500, 20.2%) and men (*n* = 5000, 19.4%). The categorical structure of the variables was taken into account by using a polychoric correlation matrix as input for the EFA [46]. To determine the number of factors, optimal coordinates and parallel analysis were used and convergence of these indicators was tested [47]. A cutoff value for factor loadings of 0.30 was used to determine to which factor a variable pertained. As potential factors were assumed to correlate, oblique rotation was applied [48].

After the EFA, CFA was performed on the remainder of the data, separately for women (*n* = 5927, 79.8%) and men (*n* = 20,733, 80.6%), to test how well the EFA-established factor structure fitted the new data. The following fit measures were used to assess the fit of the model: comparative fit index (CFI) (>0.95), root mean square error of approximation (RMSEA) (<0.06), standardized root mean residual (SRMR) (<0.08) [49]; the chi-squared test is almost always significant with large sample sizes so it was not used as a measure of fit here. Cronbach’s alphas were measured for the established factors to assess their internal consistency. With regard to the validity of the factors—used as subscales—it must be noted that, given the exploratory nature of the study, no research into divergent and convergent validity could be undertaken; also, the development of the items was not part of a validating process as the survey was already finished before the research design was set up. This means that validity of the subscales was not extensively tested and that the tentative conclusions this study proposes need to be confirmed by follow-up research.

After CFA, logistic regression analyses were conducted to assess the predictive value of the established factors. The CFA subsamples were used, separately for women and men, with as dichotomous outcome variable “Experiencing the need for help for PH” and as predictors the CFA-established factors and the covariate “Age”; variables that did not load well on any of the factors were also included as covariates to assess their predictive power for experiencing the need for help. Odds ratios (OR) with 99% confidence intervals (CI) are reported, and factors or covariates were considered significant if *p* < 0.01; this divergence from the normal alpha level of 0.05 was chosen to account for the large sample sizes and the exploratory nature of this study. Also, AUC values for the established factors were assessed to measure their power to discriminate PH from other conditions. Figures are presented showing the association between the number of indicators present for each of the factors and the probability to experience the need for help for PH. If an increase in subscale score led to a substantial increase in probability of needing help, this was taken to indicate that a measure of severity is possible and warrants further investigation. For all analyses, the open-source statistical environment R, version 3.6.1 (R Foundation for Statistical Computing, Vienna, Austria) was used with the “pROC” package for AUC calculations, the “psych” package for the EFA and the “lavaan” package for the CFA [46,50,51].

## 3. Results

### 3.1. Characteristics of Participants

Table 1 shows the characteristics of the sample subdivided into women (*n* = 7427, 22.4%) and men (*n* = 25,733, 77.8%) and into participants experiencing a need for help (*n* = 7583, 22.9%) and those not needing help (*n* = 25,577, 77.1%). Also, AUC values are reported in Table 1 to assess the power of each individual indicator to discriminate between participants experiencing the need for help and those not wanting help for PH. The AUC value below 0.5 of “Age” for women signifies here that younger women more often experience the need for help than older women. All AUC values were significantly different from 0.5 (with alpha set at 0.01) except for “Age” for men.

A larger percentage of men (25.7%) than women (12.9%) experienced a need for help for PH. Most items showed higher AUC values for women than for men, implying that these items individually better discriminated for women than for men. However, AUC values were generally similar for women and men, with the largest difference found for “Social pressure” (women: 0.64, men: 0.59) and “Less anxious” (women: 0.59, men: 0.54). The items concerning coping (except for “Need sex to function”) and “Tolerance” showed the largest differences in percentages with women endorsing these items more than men. For both genders, the item addressing “Continuation of sexual behavior despite negative consequences” had the highest AUC value and thus the highest discriminative power, respectively 0.71 for women and 0.68 for men. Typically, more than half of the sample of both women and men scored “equal to or more than one orgasm per day” on “Orgasm frequency”.

### 3.2. EFA Results

Exploratory Factor Analysis with oblique rotation yielded a four-factor structure for both women and men. In both subsamples, parallel analysis and optimal coordinates pointed to a four-factor solution. Parallel analysis is an unbiased estimator [47] and in this analysis, parallel analysis and optimal coordinates showed convergence, leading to well-interpretable four-factor solutions for women and men. The factor structure is presented in Table 2; for each of the factors, also the eigenvalues, explained variance, and Cronbach’s alpha are included in the table. In total, 52.8% of the variance was explained by the factors for women and 29.7% for men. For men, the variable “Preoccupation with sex” did not surpass the threshold of 0.30, nor did the variable “Orgasm frequency”. For women, these two variables loaded highest on the factor “Sexual Desire”. The other factor structures were the same for women and men, specifically “Negative Effects”, “Coping”, and “Extreme”. “Social pressure” showed the largest difference in loadings (on “Negative Effects”) between women and men.

### 3.3. CFA Results

Results of the CFA confirmed the EFA solution. The models for women and men were only different in the factor “Sexual Desire”, as presented in the description of EFA results. For the construction of the other factors, see Table 2 (in bold). The fit of the CFA for women was good: CFI: 0.98, RMSEA: 0.041 (95% CI: 0.040−0.043), SRMR: 0.056. Factor loadings ranged from 0.50 (“Use of drugs”) to 0.87 (“Unusual sexual stimuli”). For men, the fit values were also good: CFI: 0.96, RMSEA: 0.044 (95% CI: 0.043−0.045), SRMR: 0.057. Factor loadings ranged from 0.45 (“Use of drugs”) to 0.81 (“Continue despite negative consequences”). The value of Cronbach’s alpha for most factors—used as subscales—is questionable with values between 0.56 (“Extreme” for men) and 0.68 (“Coping” for men); only the “Coping” factor for women shows an acceptable value of 0.76. The value of 0.46 for “Sexual Desire” for men actually presents a correlation between “Need sex to function” and “Tolerance”.

### 3.4. Logistic Regression Results

Odds ratios, 99% confidence intervals and *p*-values of the factors and covariates that were used in the logistic regression are presented in Table 3.

Most notable is the high odds ratio for the factor “Negative Effects”, signifying a large effect in positively predicting the need for help for PH. “Coping” is not a significant predictor in the model for women or for men. “Extreme” is a significant positive predictor both for women and for men, suggesting that higher scores on this factor increase the probability to experience the need for help. “Sexual Desire” is significantly and negatively predictive for women and men, meaning that higher scores predict a lower probability of experiencing the need for help. For women, this means that a higher score on any of the four indicators “Need sex to function”, “Tolerance”, “Orgasm frequency”, and “Preoccupation with sex” predicts a lower probability of experiencing the need for help for PH. For men, this means that a higher score on “Need sex to function” and “Tolerance” predicts a lower probability of experiencing the need for help. “Orgasm frequency”, included as a covariate in the analysis for men, was a significant negative predictor while the covariate “Preoccupation with sex” was a significant positive predictor of experiencing the need for help for men.

### 3.5. Measure of Severity of PH

Figure 1 presents the association between each of the factors and experiencing the need for help for PH, for both women and men. For men, also the covariates “Orgasm Frequency” and “Preoccupation with sex” and their association with the need for help are presented in Figure 1 (in the “Sexual Desire” subplot). Each factor is presented with the other factors fixed at their middle score (e.g., for “Negative Effects” this is the middle of the range 0 to 6 which is 3). Especially, the association between “Negative Effects” and experiencing the need for help shows a large increase in the probability of needing help when more indicators of the factor are present, suggesting that with more indicators of “Negative Effects” present there is a substantial increase in the probability to be experiencing the need for help for PH.

The AUC values for each of the factors and covariates, presented in Table 4, suggest that “Negative Effects” is the most important factor in discriminating those experiencing the need for help from those not experiencing the need for help, both for women (AUC: 0.80) and for men (AUC: 0.78). This discriminative power can be considered acceptable to excellent [52]. The other AUC values are lower and signify poor discriminative power [52]. Note that for men “Sexual Desire” only consists of “Need sex to function” and “Tolerance”; “Orgasm frequency” and “Preoccupation with sex” are part of the “Sexual Desire” factor for women, but these indicators are analyzed as separate covariates for men.

## 4. Discussion

Main results of this study show that the “Negative Effects” factor, consisting of six indicators, is most predictive of experiencing the need for help for PH. Of this factor, we specifically want to mention “Withdrawal” (being nervous and restless) and “Loss of pleasure”. The relevance of these indicators in distinguishing PH from other conditions has been assumed [23,28] but has not previously been established by empirical research. Of the four other indicators that are part of the “Negative Effects” factor, “Fail to quit”, “Continue despite negative consequences”, and “Occurrence of negative consequences” have previously been established as predictors of PH [23,26,29] and consequently are part of all three diagnostic models of PH. The importance of “Social pressure” in association with PH has been noted [17] and possibly this characteristic is associated with moral (self-)disapproval [53]. In the Compulsive sexual behavior disorder diagnostic guidelines, it is specifically mentioned that problems due to shame and guilt are not reliably indicative of underlying disorder [28]. Careful further examination of the aspect of “Social pressure” is needed to show if it points to overpathologizing or if social pressure is caused by negative social consequences (loss of friendship, break-up [54]). Associations with moral feelings have not been tested in this study but might play a significant role in origin and continuation of PH. Given the discriminative power of the “Negative Effects” factor, this factor can be used to assess PH in a subpopulation of those in doubt about being afflicted by PH. Also, when more indicators of “Negative effects” are present, the probability of experiencing the need for help largely increases. This suggests that a measure of severity of PH could be based on the items of this factor. It needs to be mentioned that, given the low internal consistency, the development of this factor into a valid measurement instrument will need to make use of a larger number of similar items/indicators in order to measure the construct of Negative Effects better. More research is needed to establish what indicators could be added to the Negative Effects subscale to improve its internal consistency.

Results furthermore showed a grouping of five items subsumed under the “Coping” factor. These items specifically addressed post-sex effects (e.g., “I am better able to cope with daily hassles after sex”). The “Coping” factor did not significantly predict experiencing the need for help for women or for men, suggesting that “Coping” cannot be used to distinguish those experiencing the need for help from individuals not wanting help. Our exploratory study does not warrant a definite conclusion regarding the associations of coping and PH because the items concerning “Coping” were directed at post-sex effects, and sex used as coping in PH might also constitute an important aspect of initiating sex [33]. We suggest that, firstly, important prior research on PH and coping [33] is replicated and that, secondly, other associations between sex used as coping and PH are studied before definite conclusions with regard to coping and PH are drawn. Our results may explain, however, the high percentage of false positives found with the Hypersexual Behavior Inventory- [4,5], an instrument explicitly incorporating a “Coping” scale [26] to assess PH. A promising methodology to investigate sex used as coping in PH is presented by experience sampling research [55], as this type of research allows for testing the temporal structure of dysfunctional coping dynamics of individuals afflicted by PH [56].

The third factor established in our study was “Sexual Desire”, including indicators such as “Tolerance” and “Need sex to function”. “Sexual Desire” negatively predicts experiencing the need for help for both women and men. This means that when one needs sex (to function), or wants sex more and more, the probability of experiencing the need for help decreases. For women, “Sexual Desire” also includes “Preoccupation with sex” and “Orgasm frequency”. For men, these indicators were added as covariates to the analyses, and the results show that, for men, “Preoccupation with sex” is associated with a higher probability to experience the need for help while “Orgasm frequency” is associated with a lower probability to need help. These results are in line with previous research on sexual desire [37,38], but are contrary to expectations based on a sex addiction perspective. In analogy with substance use, individuals “using” certain behaviors very frequently (e.g., gambling or sex) can be expected to be at a higher risk of developing a behavioral addiction [24,57]. In the current sample, however, those participants with higher orgasm frequency were less at risk of experiencing problems with hypersexuality, from which we tentatively conclude that a cutoff between problematic and unproblematic sexual frequency [37,58] cannot be established. Likewise, “Tolerance” (wanting sex more and more) cannot be used to assess PH; as part of the “Sexual Desire” factor, it is negatively predictive of PH. This research shows that it is first and foremost the “Negative Effects” factor that indicates if hypersexuality is experienced as problematic. Increased sexual desire and higher sexual frequency are not good indicators of PH in a sample of people in doubt about their level of PH.

The last factor revealed in our data, “Extreme”, consists of four indicators addressing “Unusual sexual stimuli”, “Use drugs while watching porn”, “Extreme porn”, and “Time spent on porn”. These indicators address an escalating pattern with regard to porn-watching and paraphilic behavior. For both women and men, “Extreme” is positively predictive of the need for help. However, the discriminative power of “Extreme” is small, and in its current form this factor cannot be considered as a good indicator of PH. Further studies should be undertaken to assess the association between extreme sexual behaviors and PH.

In this sample, the percentage of men that experienced the need for help was approximately twice as large as the percentage of women. However, overall response patterns for women and men were similar in this study. We do note that gender differences on separate indicators were most prominent for “Social pressure” and coping effects; these indicators predisposed more for PH for women than for men, and further research is warranted to investigate these differences.

We would like to mention some limitations of this study: (1) PH is measured by “experiencing the need for help for PH”, a measure solely based on self-assessment that might be influenced by societal norms and thus not be an indication of underlying disorder [28]. The risk of overpathologizing one’s own hypersexual behavior due to societal norms [8,9,10,11,15] might be avoided by also including clinical assessment of participants by an experienced sexologist [16]; (2) Reliability of the subscales is generally not high, which means that caution should be observed when interpreting the exploratory results of this study; follow-up research should focus on the development of extended subscales with items that are more aligned in order to arrive at better internal consistency; importantly, such research might also improve discriminative power of subscales (though already sufficiently high in the case of “Negative Effects”); (3) Heterogeneity of the sample with regard to the sexual behavior associated with PH (e.g., porn addiction or compulsive cheating) might have confounded results and needs to be taken into account in further research; (4) The sample, though large, consisted of self-selected respondents who did not clarify their reasons for participation. However, the high number of women and men in this study who experience the need for help for PH, and the introduction to the survey that clearly states its purpose of providing a preliminary assessment of sex addiction, suggests that responses have indeed been sampled of those in doubt about their level of PH; (5) This research did not differentiate gender beyond the woman–man dichotomy; follow-up research needs to consider including a more differentiated measure of gender identity; and (6) Comorbidity has not been investigated in this research while, on the other hand, it is known to be a common factor of PH (e.g., in bipolar disorder) [59] that should be taken into account in follow-up research.

Despite the limitations mentioned, we think that this research contributes to the field of PH research and to the exploration of new perspectives on (problematic) hypersexual behavior in society. We stress that our research showed that “Withdrawal” and “Loss of pleasure”, as part of the “Negative Effects” factor, can be important indicators of PH. On the other hand, “Orgasm frequency”, as part of the “Sexual Desire” factor (for women) or as a covariate (for men), did not show discriminative power to distinguish PH from other conditions. These results suggest that for the experience of problems with hypersexuality, attention should focus more on “Withdrawal”, “Loss of pleasure”, and other “Negative Effects” of hypersexuality, and not so much on sexual frequency or “excessive sexual drive” [60] because it is mainly the “Negative Effects” that are associated with experiencing hypersexuality as problematic. Based on the current research, we recommend to incorporate items addressing these characteristics in a measurement instrument for PH. This would mean that characteristics from different diagnostic models should be integrated into one instrument [14]. Theoretically, this would suggest that a comprehensive integration of current conceptualizations of PH is expedient that takes the unique nature of problematic hypersexuality into account in relation to societal norms and physical and mental well-being.

## 5. Conclusions

This exploratory research suggests that the factor “Negative Effects” will be the most optimal in correctly assessing PH and discriminating PH from other conditions. To this factor belong, amongst others, the indicators of “Withdrawal” and “Loss of pleasure” that in the past were solely attributed to one of the three diagnostic models for PH. With respect to theory, this implies that PH possibly should not be classified under existing conceptualizations as an addictive, hypersexual, or compulsive sexual behavior disorder but might better be viewed from a theoretically more comprehensive perspective. With respect to clinical practice, the results of this study suggest that the empirically relevant indicators used in different diagnostic models to assess PH might best be joined to construct an instrument to assess presence and severity of PH. Future research to develop and validate such an instrument should specifically be undertaken in the same relevant subpopulations as where it will be applied, to avoid overpathologizing of nonproblematic sexual behavior. Gender differences in PH need to be considered, and instruments to assess PH should at least partly be different for women and men.

## Figures and Tables

**Figure 1 ijerph-17-06907-f001:**
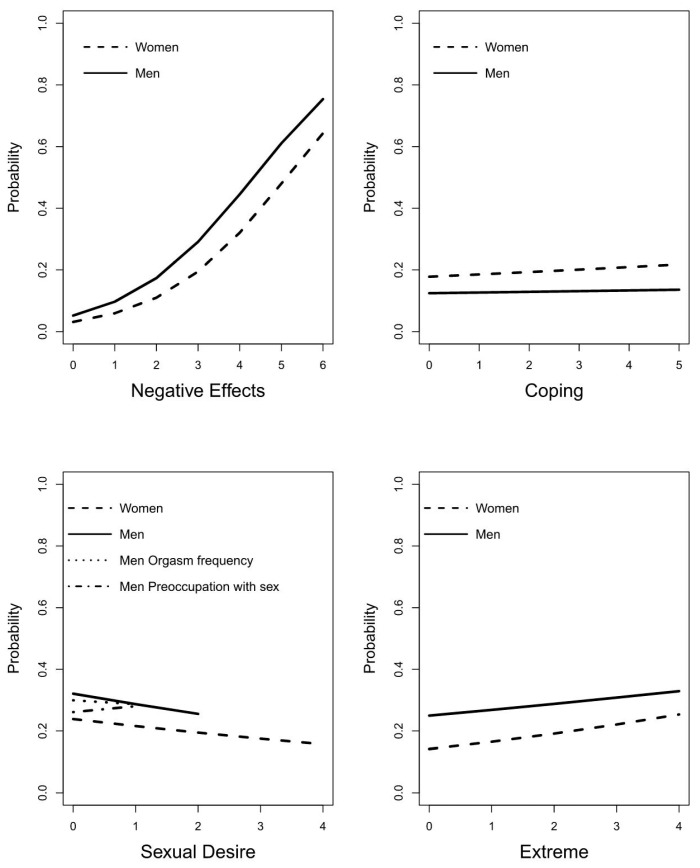
Association between number of indicators present for each factor (and two covariates for men) and the probability of experiencing the need for help for PH.

**Table 1 ijerph-17-06907-t001:** Description of the sample accounting for gender and experiencing the need for help for problematic hypersexuality (PH).

Indicator Variables and Covariates	Experiences the Need for Help for PH. Women: *n* (%) (of a Total of 958) Men: *n* (%) (of a Total of 6625)	Does not Want Help for PH. Women: *n* (%) (of a Total of 6469) Men: *n* (%) (of a Total of 19,108)	AUC Women Men
Preoccupation with sex	611 (63.8%) 4736 (71.5%)	2827 (43.7%) 9700 (50.8%)	0.60 0.60
Failed to quit	696 (72.6%) 5401 (81.5%)	2428 (37.5%) 9232 (48.3%)	0.68 0.67
Negative consequences	478 (49.9%) 3826 (57.7%)	1223 (18.9%) 5205 (27.2%)	0.66 0.65
Continue despite negative consequences	759 (79.2%) 5704 (86.1%)	2392 (37.0%) 9668 (50.6%)	0.71 0.68
Tolerance	691 (72.1%) 3439 (51.9%)	3908 (60.4%) 7702 (40.3%)	0.56 0.56
Withdrawal (range: 0–4), mean (SD)	1.92 (1.34) 1.78 (1.19)	1.08 (1.25) 1.14 (1.19)	0.68 0.66
Need sex to function	631 (65.9%) 3615 (54.6%)	3369 (52.1%) 9277 (48.6%)	0.57 0.53
Distracted by sex	679 (70.9%) 3914 (59.1%)	3982 (61.6%) 9503 (49.7%)	0.55 0.55
Feel stronger	454 (47.4%) 1893 (28.6%)	2376 (36.7%) 4939 (25.8%)	0.55 0.51
Less depressed	502 (52.4%) 2479 (37.4%)	2386 (36.9%) 5492 (28.7%)	0.58 0.54
Less anxious	390 (40.7%) 1493 (22.5%)	1530 (23.7%) 2526 (13.2%)	0.59 0.54
Better deal with life	407 (42.5%) 1626 (24.5%)	2131 (32.9%) 4274 (22.4%)	0.55 0.51
Loss of pleasure	513 (53.5%) 3958 (59.7%)	1496 (23.1%) 6035 (31.6%)	0.65 0.64
Orgasm frequency	529 (55.2%) 4174 (63.0%)	3368 (52.1%) 11,858 (62.1%)	0.53 0.52
Time spent on porn (hours), mean (SD)	21 min (20 min) 42 min (37 min)	15 min (17 min) 32 min (33 min)	0.59 0.58
Extreme porn (range: 0–3), mean (SD)	2.02 (1.12) 2.22 (0.77)	1.70 (1.16) 2.09 (0.79)	0.58 0.55
Use drugs while watching porn (range: 0–4), mean (SD)	1.43 (0.87) 1.34 (0.72)	1.29 (0.76) 1.30 (0.68)	0.55 0.51
Social pressure	423 (44.2%) 2136 (32.2%)	1006 (15.6%) 2760 (14.2%)	0.64 0.59
Unusual sexual stimuli (range: 0–4), mean (SD)	0.51 (0.96) 0.37 (0.77)	0.28 (0.71) 0.23 (0.61)	0.56 0.54
Age, mean (SD)	31 years 6 months (8 years y 11 months) 36 years 2 months (11 years 8 months)	32 years 4 months (9 years 4 months) 36 years 3 months (12 years 4 months)	0.47 0.50

**Table 2 ijerph-17-06907-t002:** Factor loadings of the variables in the exploratory factor analysis (EFA). Indicators with factor loadings in bold pertain to the factor column they are in.

Potential Indicators of PH	Negative Effects Women/Men	Coping Women/Men	Extreme Women/Men	Sexual Desire Women/Men
Fail to quit	0.69/0.61			
Negative consequences	0.65/0.43			
Continue despite negative effects	0.86/0.69			
Loss of pleasure	0.55/0.51			
Social pressure	0.75/0.31			
Withdrawal	0.51/0.44			
Distracted by sex		0.68/0.44		
Feel stronger		0.76/0.41		
Less depressed		0.83/0.68		
Less anxious		0.90/0.62		
Better deal with life		0.61/0.39		
Extreme porn			0.80/0.69	
Time spent on porn			0.84/0.60	
Use drugs while watching porn			0.38/0.30	
Unusual sexual stimuli			0.39/0.35	
Need sex to function				0.70/0.56
Tolerance				0.52/0.39
Preoccupation with sex				0.41/0.29
Orgasm frequency				0.47/0.22
Explained variance	16.8%/9.6%	15.6%/7.9%	10.9%/6.7%	9.4%/5.5%
Total explained variance		Women: 52.8%	Men: 29.7%	
Eigenvalue	3.19/1.82	2.97/1.49	2.01/1.28	1.79/1.05
Cronbach’s alpha	0.64/0.62	0.76/0.68	0.64/0.56	0.61/0.46

**Table 3 ijerph-17-06907-t003:** Results of the logistic regression using “Experiencing the need for help” as criterion variable.

Factors/Covariates (Range)	Women OR (99% CI)	Women *p*-Value	Men OR (99% CI)	Men *p*-Value
Intercept	0.03 (0.02–0.04)	<0.001	0.05 (0.04–0.06)	<0.001
Negative Effects (0–6)	1.95 (1.84–2.10)	<0.001	1.95 (1.88–2.01)	<0.001
Coping (0–5)	1.05 (0.98–1.12)	0.066	1.02 (0.99–1.05)	0.100
Extreme (0–4)	1.20 (1.02–1.41)	0.003	1.10 (1.01–1.21)	0.005
Sexual Desire (0–4/0–2)	0.87 (0.79–0.97)	<0.001	0.85 (0.80–0.91)	<0.001
Preoccupation with sex (0–1)			1.32 (1.18–1.46)	<0.001
Orgasm frequency (0–1)			0.89 (0.80–0.99)	<0.001
Age (0–6)	1.02 (0.89–1.14)	0.735	1.02 (0.98–1.06)	0.156

**Table 4 ijerph-17-06907-t004:** AUC values and 99% confidence intervals of factors and covariates for experiencing the need for help for PH.

Factors/Covariates	Women AUC (99% CI)	Men AUC (99% CI)
Negative Effects	0.80 (0.79–0.83)	0.78 (0.77–0.78)
Coping	0.60 (0.59–0.62)	0.57 (0.56–0.58)
Extreme	0.60 (0.58–0.62)	0.58 (0.57–0.59)
Sexual Desire	0.61 (0.59–0.63)	0.56 (0.55–0.56)
Orgasm frequency (men)		0.51 (0.50–0.51)
Preoccupation with sex (men)		0.60 (0.60–0.61)
Age	0.47 (0.46–0.49)	0.50 (0.49–0.51)

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
