# Peer review of "Three Diagnoses for Problematic Hypersexuality; Which Criteria Predict Help-Seeking Behavior?"

_ijerph, 2020, doi:10.3390/ijerph17186907_

Round 1
Reviewer 1 Report
This is a nice paper on the assessment of problematic hypersexuality. Background provides adequate information, results are clearly presented and discussion is interesting.
However, I have the following observations:
-gender: in a paper on problematic sexuality why did you include only male and female participants? How about non-binary participants, i.e. trangender people?
-in your sample there is a disproportion between male and female participants. Could you please discuss the risk of bias?
-you did not control your results for manic symptoms. Since hypersexuality could be a symptom of mania this issue should be discussed a bit.
Author Response
Dear reviewer,
Thank you for your review and attention. Our adaptations to the text and response to your comments can be found below:
Review Report Form
This is a nice paper on the assessment of problematic hypersexuality. Background provides adequate information, results are clearly presented and discussion is interesting.
However, I have the following observations:
-gender: in a paper on problematic sexuality why did you include only male and female participants? How about non-binary participants, i.e. transgender people?
The reason we have not included non-binary participants is that our analyses are based on secondary data that was collected before our research design was implemented. Had we been able to direct data collection, we would have included non-binary participants. The characterization of the data as secondary is more elaborated upon in the revised version of the article (section 2.3, newly added). Also, the following sentence has been added to the limitations of this study (Discussion):
“4) This research did not differentiate gender beyond the woman-man dichotomy; follow-up research needs to consider including a more differentiated measure of gender […]” (lines 413-415)
-in your sample there is a disproportion between male and female participants. Could you please discuss the risk of bias?
In order to prevent the risk of bias due to gender, we have executed separate analyses for women and men; we did not specifically mention that this prevents the risk of bias due to gender and have done that in the revised version of the article in the section 2.10. “Statistical Analyses”:
“Analyses were conducted separately for women and men as response patterns for both sexes are expected to be different and we wanted to investigate these differences; using separate analyses also avoids the risk of bias of gender.” (lines 193-195)
-you did not control your results for manic symptoms. Since hypersexuality could be a symptom of mania this issue should be discussed a bit.
The subject of mania as a possible reason for hypersexuality, touches upon the subject of comorbidity and PH in general. We mention this aspect in the limitations of the study; due to the post-hoc research design (see section 2.3, newly added) we have not been able to study comorbidity:
“[…] and 5) Comorbidity has not been investigated in this research while on the other hand it is known to be a common factor of PH (e.g. in bipolar disorder) [59]. We suggest that this aspect is taken into account in follow-up research.” (lines 415-417)
Submission Date
14 August 2020
Date of this review
07 Sep 2020 16:31:21

Reviewer 2 Report
This research has proposed some highly interesting proposition on sexual addiction. I would like to post some comments /questions :
Ethical consideration
The dataset has been published in Dutch language before (Van Tuijl, P., Tamminga, A., Meerkerk, G-J., Verboon, P., Leontjevas, R,van Lankveld, J. (2019). Het seksverslavingsmodel van hyperseksualiteit getoetst: essentiële kenmerken en hun samenhang met hulpbehoefte, Tijdschrift voor Seksuologie, 43(4) [845].
Ethical approval number was not given
parental approval for use of survey data were not taken for participants at 17 years old
Caption for Table 4 in ambiguous ( AUC values for the factors and covariates. Outcome: experiencing the need for help for PH (y/n).
Some info of Figure 1 caption should be placed at Method instead: Orgasm frequency” and “Preoccupation with sex” for men (both range: 0 – 1) were analyzed as covariates and are included in 274 the “Sexual Desire” subplot.
aim of this study was to
-test an extended set of characteristics specified as criteria in the three different diagnostic models for PH-this dataset were presented before
-did you explain the diagnostic model?
-investigate if a larger number of endorsed indicators increases the odds of experiencing the need for help for PH
-not clear , pls explain endorsed indicator
However, it seems that these are not well summarized in the conclusion section.
This research suggests that the factor “Negative effects”, with indicators stemming from different diagnostic models, will be the most optimal in correctly assessing PH and in discriminating PH from other conditions.
-which diagnostic model?
With respect to theory, this implies that PH possibly should not be classified under existing conceptualizations as an addictive, hypersexual or compulsive behavior disorder but might better be viewed from a theoretically more comprehensive perspective.
-is it proven statistically in this study?
With respect to clinical practice, the results of this study suggest that the empirically relevant indicators used in different diagnostic models to assess PH, might best be joined to construct an instrument to assess presence and severity of PH.
-has it been tested using the method described?
Language errors are highlighted pls refer to attachment

Author Response
Dear reviewer,
Thank you for reviewing our article, we have taken to heart your suggestions which, we hope, is visible in our revision of the article.
Herewith our reactions to your review:
(Comments of the authors in italics.)
Ethical consideration
The dataset has been published in Dutch language before (Van Tuijl, P., Tamminga, A., Meerkerk, G-J., Verboon, P., Leontjevas, R,van Lankveld, J. (2019). Het seksverslavingsmodel van hyperseksualiteit getoetst: essentiële kenmerken en hun samenhang met hulpbehoefte, Tijdschrift voor Seksuologie, 43(4) [845].
The nature of the analyses and the article itself are different from the previous article. The previous article was written in Dutch and did not look at the data from the perspective of the three different diagnostic models. Also, the analyses were different as there was no Factor Analysis included and also no separate analyses for women and men. Consequently, it did not reach the conclusion that a more comprehensive perspective on Problematic Hypersexuality is expedient.
Ethical approval number was not given
As the data were considered secondary data by the ethical committee, ethical approval was waived. The documents supporting this, have been sent to the editors. Changes in the text of the article have been made to state this more clearly:
“The first objective of the survey was to provide feedback on the level of PH of participants completing the survey. Before and after taking the survey, participants were notified that the data collected might also be used for scientific research. Data was not collected with a research strategy in mind and the current research has been set up after data collection finished. As the data was classified as secondary data, it was considered exempt from ethical approval by the ethical approval board of the Open University Netherlands. In order to secure anonymity, IP addresses were not recorded but changed into anonymous code. Completed surveys did not contain information that could be traced back to participants.” (lines 102-109)
parental approval for use of survey data were not taken for participants at 17 years old
Data of participants younger than 22 have not been used. This is expressed in the article by the following (revised) text:
“Completed surveys were excluded when: 1) participants belonged to the age group 17-21 or younger (n = 17,689) because parental approval could not be obtained;” (line 110)
From the age of 18 and upward parental approval is not needed; participants were not asked about their exact age but only to what age group they belonged. As the age group containing ages younger than 18 (i.e. 17) also contained responses of those from 18 to 21, the whole age group of 17 to 21 years of age was excluded from the current research.
Caption for Table 4 in ambiguous ( AUC values for the factors and covariates. Outcome: experiencing the need for help for PH (y/n).
The caption of Table 4 has been replaced by: “AUC values of factors and covariates for experiencing the need for help for PH.”
Some info of Figure 1 caption should be placed at Method instead: Orgasm frequency” and “Preoccupation with sex” for men (both range: 0 – 1) were analyzed as covariates and are included in 274 the “Sexual Desire” subplot.
This information has been taken out of the caption of Figure 1 and has been placed in the text more clearly:
“For men also the covariates “Orgasm Frequency” and “Preoccupation with sex” and their association with the need for help, are presented in Figure 1 (in the “Sexual Desire” subplot).” (lines 310-312)
It has been placed in the Result section instead of the Method section because “Orgasm frequency” and “Preoccupation with sex” only after analyses proved to be covariates according to the rules described in the Method section (“[..] variables that did not load well on any of the factors were also included as covariates to assess their predictive power for experiencing the need for help […]”) (lines 228-229).
aim of this study was to
-test an extended set of characteristics specified as criteria in the three different diagnostic models for PH-this dataset were presented before
-did you explain the diagnostic model?
The three different diagnostic models are presented in the Introduction. Also, it has been stated more clearly in the current version of the article that this research does not aim to provide an alternative diagnostic model but offers preliminary investigations for further research into a more comprehensive diagnostic model. The reason we cannot make definite claims also has to do with the exploratory nature of this research. This is explained in more detail in section 2.3, a section added to the revised article that considers validity issues of this exploratory research.
-investigate if a larger number of endorsed indicators increases the odds of experiencing the need for help for PH
-not clear , pls explain endorsed indicator
With endorsed indicators is meant those indicators that are acknowledged to be present by the participants. This expression has been changed to “indicators present”.
However, it seems that these are not well summarized in the conclusion section.
This research suggests that the factor “Negative effects”, with indicators stemming from different diagnostic models, will be the most optimal in correctly assessing PH and in discriminating PH from other conditions.
-which diagnostic model?
This research shows that none of the three currently used diagnostic models is the best diagnostic model for the “behavioral complex” of Problematic Hypersexuality. It is suggested that if a diagnostic model is to be developed, indicators from different diagnostic models should be included in it.
Added to the conclusion is the following sentence, mentioning those indicators of the “Negative Effects” factor that show the inclusion of criteria from different diagnostic models:
“To this factor belong amongst others the indicators of “Withdrawal” and “Loss of pleasure” that in the past were solely attributed to one of the three diagnostic models for PH.” (lines 435-437)
With respect to theory, this implies that PH possibly should not be classified under existing conceptualizations as an addictive, hypersexual or compulsive behavior disorder but might better be viewed from a theoretically more comprehensive perspective.
-is it proven statistically in this study?
The statement is not proven by the current research but only suggested. To prove the theory that proposes a more comprehensive perspective, will take a larger amount of research. Also, these analyses were exploratory and thus not well equipped to reach definite conclusions in this regard. With the statement “With respect to theory […]” we aimed to direct future research towards the more comprehensive perspective that has been suggested (but not proven) by this study. See also the newly added section 2.3 considering the explorative nature of this research.
With respect to clinical practice, the results of this study suggest that the empirically relevant indicators used in different diagnostic models to assess PH, might best be joined to construct an instrument to assess presence and severity of PH.
-has it been tested using the method described?
No, it has not been tested with the method described, but rather it has been explored by the method described. The current research must be considered exploratory research rather than hypothesis testing research. This is a consequence of the fact that we made use of secondary data and a research design that was set up after data collection had finished. This made our analyses explorative instead of confirmative. However, our explorative research does suggest that an instrument to assess presence and severity of PH best consists of indicators from different diagnostic models. This needs to be confirmed by follow-up research. The explorative nature of our research has been stressed more and incorporated in the text specifically in section 2.3, a newly added section in the revised article.
Language errors are highlighted pls refer to attachment
The highlighted errors have been noted and improvements made; also, a number of other textual improvements have been executed (visible by “track changes”).
Submission Date
14 August 2020
Date of this review
28 Aug 2020 08:49:27

Reviewer 3 Report
The authors investigated what factors could serve as indicators of problematic hypersexuality in a large survey of 58,158 participants. Among other factors, they found that across both genders, the ‘negative effects’ factor and ‘sexual desire’ factors were respectively positively and negatively predictive of problematic hypersexuality.
I feel that this is a promising paper but the writing is extremely confusing and difficult to follow. A major revision is necessary to rewrite and reorganize the manuscript. I have several suggestions to improve the manuscript:
1. First, the paper would benefit from proofreading for minor improvements to the flow of some sentences as well as for grammar; some sentences would benefit from additional commas to ease understanding (e.g. “…were used as diagnostic criteria for sex addiction a high prevalence was found…” in line 61 would be more easily read with a comma after “sex addition”), while some sentences are ungrammatical due to extra commas (e.g. “…in the one study investigating this, a comparison group that was not affected by PH, was not included” in lines 62–32 has an extra comma after “PH”). An example of incorrect grammar is in line 163, “analyses were be performed”. I would also request that the authors refrain from using too many acronyms (e.g. CSBD, ROC, OC, PA) as it is not easy for the reader to remember all of them.
2. The introduction is clear and the rationale for the study is well-explained, but it would be much easier to read if it was split into two or more paragraphs.
3. The methods are somewhat unclear, particularly lines 105–115.
4. Descriptive statistics on participants' characteristics should be elaborated further in the method section
5. I am not sure what the reader is supposed to understand from being told that “the number of people … who consider themselves to be fairly or severely addicted to porn was 85,000” (which does not seem to be referring to the current sample) which is immediately followed by information about the current sample (which is relevant to the current sample).
6. Additionally, while the rationale for choosing the subpopulation is adequately explained in line 115, it is unclear how exactly this subpopulation is determined.
7. Reliability of the indicators of problematic hypersexuality, sex addition, compulsive sexual behaviour disorder and hypersexual disorder must reported
8. More clarification is necessary related to the validity of the scales used in the study.
9. More justification should be provided regarding the covariates that were used in the analysis. The inclusion of these covariates should be justified further
10. Importantly, In subsection 2.7 Need for help, I disagree with the authors that these items are sufficiently indicative of a need for help—a respondent could feel that they *need* help but at the same time not *want* any help. Both items are worded to assess the *want* for help (“I would like…”).
11. The statistical analyses are difficult to follow and unclear. The purpose of each statistical analyses in relation to the goal of the study should be explained clearly. The authors should consider to use subheading to organize their method and results better.
12. Lastly, similar to the introduction, the discussion section can be more succinct, or otherwise split into more paragraphs to allow easier reading.
Author Response
Dear Reviewer,
Thank you for your attention and review. We have adapted our text and taken to heart your suggestions. Herewith our response and adaptations:
Comments by the authors are in italics.
The authors investigated what factors could serve as indicators of problematic hypersexuality in a large survey of 58,158 participants. Among other factors, they found that across both genders, the ‘negative effects’ factor and ‘sexual desire’ factors were respectively positively and negatively predictive of problematic hypersexuality.
I feel that this is a promising paper but the writing is extremely confusing and difficult to follow. A major revision is necessary to rewrite and reorganize the manuscript. I have several suggestions to improve the manuscript:
- First, the paper would benefit from proofreading for minor improvements to the flow of some sentences as well as for grammar; some sentences would benefit from additional commas to ease understanding (e.g. “…were used as diagnostic criteria for sex addiction a high prevalence was found…” in line 61 would be more easily read with a comma after “sex addition”), while some sentences are ungrammatical due to extra commas (e.g. “…in the one study investigating this, a comparison group that was not affected by PH, was not included” in lines 62–32 has an extra comma after “PH”). An example of incorrect grammar is in line 163, “analyses were be performed”. I would also request that the authors refrain from using too many acronyms (e.g. CSBD, ROC, OC, PA) as it is not easy for the reader to remember all of them.
Extensive proofreading has been undertaken which has led to improvements, we hope. Often a simplification of sentences was possible and has been executed, hopefully rendering the text more readable.
Less acronyms have been used. For SA (Sex Addiction), HD (Hypersexual Disorder), CSBD (Compulsive Sexual Behavior Disorder), OC (Optimal Coordinates), PA (Parallel Analysis), ROC (Receiver Operating Curve) the acronyms have been replaced by the full wording. In some cases the acronyms have been maintained (specifically for CFI, RSMR, RMSEA, AUC, PH, CFA,EFA,OR and CI). This is done to avoid extensive repetition of expressions. Most of the acronyms used are more generally used in research articles and can be assumed known (e.g. CFA or AUC).
- The introduction is clear and the rationale for the study is well-explained, but it would be much easier to read if it was split into two or more paragraphs.
This has been done now.
- The methods are somewhat unclear, particularly lines 105–115.
These lines have been revised and the mention of the porn addiction research has been taken out of the text.
- Descriptive statistics on participants' characteristics should be elaborated further in the method section
To section “2.2. Survey and sample” the following lines have been added:
(In total 33,160 completed surveys were included in the analyses) “of which 25,733 (77.8%) were filled in by men and 7,427 (22.4%) by women. In total 7,583 (22.9%) participants expressed interest in seeking help for PH.” (lines 113-114)
- I am not sure what the reader is supposed to understand from being told that “the number of people … who consider themselves to be fairly or severely addicted to porn was 85,000” (which does not seem to be referring to the current sample) which is immediately followed by information about the current sample (which is relevant to the current sample).
The reference to the porn addiction research has been taken out of the article as porn addiction and sex addiction are indeed two different concepts.
- Additionally, while the rationale for choosing the subpopulation is adequately explained in line 115, it is unclear how exactly this subpopulation is determined.
The following sentence has been added to the newly added section 2.3 (about the exploratory nature of this research):
“Though there is no certainty that the investigated subpopulation consists indeed of those in doubt about their level of sex addiction, the introduction to the survey stresses its purpose as providing a preliminary self-assessment of the participant’s level of sex addiction; also completion of the survey, needed to receive the feedback, shows an interest in the outcome of the survey and suggests the targeted subpopulation was reached.” (lines 131-135)
- Reliability of the indicators of problematic hypersexuality, sex addition, compulsive sexual behaviour disorder and hypersexual disorder must reported
The items/indicators of the different diagnostic models are not the complete set of indicators for those diagnostic models and thus it would not lead to proper results to determine reliability (e.g. with Cronbach’s alpha) for sex addiction or hypersexual disorder; for compulsive sexual behavior disorder only one unique indicator was present in the survey; for problematic hypersexuality the set of indicators only just now with this research has been explored. It could be argued that the “Negative Effects” factor is the scale to assess PH but this needs to be confirmed with follow-up research and the development of a measurement instrument. This study is only a first explorative step in that direction. The reliability of the factors (i.e. Cronbach’s alpha) has been added in the results section, particularly to Table 2, the last line of the table.
In the Discussion section where the Negative Effects subscale is discussed, the following lines regarding its internal consistency have been added:
“It needs to be mentioned that, given the low internal consistency, the development of this factor into a valid measurement instrument will need to make use of a larger number of similar items/indicators in order to measure the construct of Negative Effects better. More research is needed to establish what indicators could be added to the Negative Effects subscale to improve its internal consistency.” (line 351 – 355)
- More clarification is necessary related to the validity of the scales used in the study.
Issues related to the validity of this research are considered in the newly added section 2.3. The validity of the scales, the four factors determined with Factor Analyses, is discussed more in section 2.10. after explaining the construction of the subscales:
“With regard to the validity of the factors - used as subscales - it must be noted that, given the exploratory nature of the study, no research into divergent and convergent validity could be undertaken; also the development of the items was not part of a validating process as the survey was already finished before the research design was set up. This means that validity of the subscales is not extensively tested and that the tentative conclusions this study proposes need to be confirmed by follow-up research.” (lines 219-224)
- More justification should be provided regarding the covariates that were used in the analysis. The inclusion of these covariates should be justified further
As explicated in the newly added section 2.3. the survey has not been designed by the researchers. This means that the researchers have not chosen the items (and potential covariates) included in the survey; a choice could be made and has been made with regard to the variables included in the analyses. This choice has been elaborated upon in section 2.9:
“With regard to “Orgasm frequency”, “Time spent on porn” and “Unusual sexual stimuli” (paraphilic orientation) prior research showed some associations with PH but these associations remained equivocal. With regard to “Watch porn while using drugs” and “Extreme porn” there has been less research but these two indicators would align with an escalating pattern of PH as conceptualized in the sex addiction perspective and thus were included as covariates. “Social Pressure” has also not been investigated quantitatively but has been suggested as an equivocal aspect of PH by therapists [17] that needs to be studied more.” (lines 175-182)
- Importantly, In subsection 2.7 Need for help, I disagree with the authors that these items are sufficiently indicative of a need for help—a respondent could feel that they *need* help but at the same time not *want* any help. Both items are worded to assess the *want* for help (“I would like…”).
We have changed the expression “In need of help […]” to “Experiencing the need for help […]”:
“Experiencing the need for help for problems due to PH”. (line 160)
It indeed remains conjecture if a participant that admits to like to have help, really needs help. In follow-up this distinction should be made between “I feel I need help for etc.” and “I would like to receive help for etc.”. This is considered a limitation of the study and addressed partly in limitation 1 of the Discussion section:
“PH is measured by “experiencing the need for help for PH”, a measure solely based on self-assessment […]” (line 401)
The solution proposed there, to also take a clinical assessment of participants, will of course not always be possible. We are aware though of the difference between what participants want and what they need (or should need, given self-awareness); in this research the responses to the two items used, are the only way available to assess if a participant experiences a need for help.
- The statistical analyses are difficult to follow and unclear. The purpose of each statistical analyses in relation to the goal of the study should be explained clearly. The authors should consider to use subheading to organize their method and results better.
We have addressed this comment in different ways. First, section 2.3. has been added to the revised article, more clearly stating that these analyses are exploratory and that the research design has been set up after data collection. The aim of the study (to explore and test a number of indicators of PH) has been extended upon by describing the aim of the analyses in the following newly added paragraph to the “2.10. Statistical Analyses” section:
“Exploratory analyses have been designed to investigate the indicators of PH available in the collected data. Special attention was given to the factor structure of the variables; establishing which indicators belonged together allowed for a better interpretation of the indicators and also made it possible to investigate the predictive and discriminating properties of the established factors. Follow-up research will be needed to confirm the exploratory results of this study.” (lines 188-192)
This paragraph explains the reasons to use the chosen analyses, we hope. What we sought to convey was that analyses were chosen (after data collection) in order to bring out as good as possible the properties of the indicators of PH available in the data. Also, we hoped to show the idea behind the “line-up” of the analyses: we first needed to have established a confirmed factor structure in order to execute meaningful regression and Receiver Operating Curve analyses. In the relevant paragraphs of section 2.10. we also added the reason for the particular analysis as well (e.g. “[…] logistic regression analyses were conducted to assess the predictive value of the established factors […]”) (line 225-226).
- Lastly, similar to the introduction, the discussion section can be more succinct, or otherwise split into more paragraphs to allow easier reading.
The Discussion section has been split into more paragraphs.
Submission Date
14 August 2020
Date of this review
02 Sep 2020 20:38:42

Round 2
Reviewer 2 Report
The comments have been considered and taken into account in the current revision.
Author Response
Dear Reviewer,
Thank you for your attention and comments!
Kind regards, Piet van Tuijl and coautors
Reviewer 3 Report
The authors have sufficiently addressed my comments. Congratulation!
But I need the authors to elaborate the lack of internal consistency/reliability in their scales in their limitation and also provide more caution interpretation of their results given the psychometric properties of their measures.
Author Response
Dear reviewer,
Thank you for your attention and comments. Herewith included our response to your comments.
Kind regards, Piet van Tuijl and coauthors
